# On Frequency Estimation and Detection of Heavy Hitters in Data Streams

**Federica Ventruto**, **Marco Pulimeno**, **Massimo Cafaro \*** and **Italo Epicoco \***

Dpartment of Engineering for Innovation, University of Salento, 73100 Lecce, Italy;
federica.ventruto@studenti.unisalento.it (F.V.); marco.pulimeno@unisalento.it (M.P.)
**\*** Correspondence: massimo.cafaro@unisalento.it (M.C.); italo.epicoco@unisalento.it (I.E.)

**Abstract:** A stream can be thought of as a very large set of data, sometimes even infinite, which arrives sequentially and must be processed without the possibility of being stored. In fact, the memory available to the algorithm is limited and it is not possible to store the whole stream of data which is instead scanned upon arrival and summarized through a succinct data structure in order to maintain only the information of interest. Two of the main tasks related to data stream processing are frequency estimation and heavy hitter detection. The frequency estimation problem requires estimating the frequency of each item, that is the number of times or the weight with which each appears in the stream, while heavy hitter detection means the detection of all those items with a frequency higher than a fixed threshold. In this work we design and analyze ACMSS, an algorithm for frequency estimation and heavy hitter detection, and compare it against the state of the art ASKETCH algorithm. We show that, given the same budgeted amount of memory, for the task of frequency estimation our algorithm outperforms ASKETCH with regard to accuracy. Furthermore, we show that, under the assumptions stated by its authors, ASKETCH may not be able to report all of the heavy hitters whilst ACMSS will provide with high probability the full list of heavy hitters.

**Keywords:** data stream mining; heavy hitters; frequency estimation; sketches

## 1. Introduction

In the data stream model, data arrives or can be accessed only sequentially and in a given order; no random access to the data is allowed. This is the reason why we refer to the input data as a stream. While a stream can be defined as a continuous and unbounded sequence of items, without loss of generality, we a priori set its length for convenience.

The space available to the algorithm is not enough to store all the data, thus only a single scan of the data is possible, and each data item has to be processed and then discarded. The model naturally applies to all data generated at massive volumes as a sequence of elements, such as the values measured by a sensor, or the TCP/IP packets that traverse a router in a network.

A streaming algorithm maintains a summary (or synopsis) data structure in order to process the stream. The summary is updated accordingly at each item arrival and requires a bounded amount of memory, much smaller than that necessary for storing the entire stream. Queries about the data stream are answered using that summary, and the time for processing an item and computing the answer to a given query is limited.

Two of the most important and well studied problems in the field of Data Mining are frequency estimation of data stream items and the detection of heavy hitters, also known as frequent items. Informally, given a stream of length $n$ consisting of pairs (item, weight) the frequency of an item is the sum of its weights values; when the weight of all of the items is equal to one (a common case), the frequency of an item coincides with the number of its occurrences within

the stream. Similarly, frequent items can be informally defined as those items in the stream whose frequency exceeds a user's defined support threshold.

Mining frequent items is also referred to in the literature as market basket analysis [1], hot list analysis [2] and iceberg query [3,4].

Determining frequent items in a stream is a problem important both from a theoretical perspective and for its many practical applications; a few examples follows: (i) Popular products—the stream may be the page views of products on the web site of an online retailer; frequent items are then the most frequently viewed products; (ii) popular search queries—the stream may consist of all of the searches on a search engine; frequent items are then the searches made most often; (iii) TCP flows—the stream may consist of the data packets passing through a network switch, each annotated with a source-destination pair of IP addresses, and the frequent items are then the flows that are sending the most traffic.

Additional applications include, for instance, analysis of web logs [5], Computational and theoretical Linguistics [6] and the analysis of network traffic [7–9].

In this paper we are concerned with the problems of frequency estimation and frequent item detection in data streams. In particular, we tackle these problems designing and analyzing ACMSS, a novel algorithm solving both problems. We compare our algorithm to ASKETCH [10], the state of the art algorithm for these problems and, through extensive experimental results, we show that ACMSS achieves better accuracy than ASKETCH for the problem of frequency estimation.

The design features leading to the improved accuracy are (i) the use of a sketch based on a space optimized version of the CMSS sketch [11], (ii) a different sketch update policy called conservative update [12,13] which is not used neither in CMSS nor in ASKETCH and (iii) a different swap policy to determine which items must be moved from the filter data structure to the sketch. Moreover, we prove that our algorithm is able to retrieve with high probability all of the frequent items, owing to the specific design of our sketch data structure, whilst ASKETCH may not be able in all of the cases to retrieve all of the frequent items.

Our ACMSS algorithm exhibits a tradeoff between accuracy and speed. Indeed, the experimental results show that ACMSS provides better accuracy than ASKETCH but at the expense of being slightly slower. However, in different application fields it is preferable being more accurate (albeit a little slower) rather than being faster. Here we recall two different application fields in which accuracy is a strict requirement: Healthcare and security. In healthcare applications [14] accuracy is of paramount importance; in mission critical applications the human life may be at risk. Regarding security, anomaly detection [15,16] is one of the most studied topics. Accurately determining the so-called outliers is crucial for many applications e.g., DDoS (Distributed Denial of Service) [17].

The rest of this paper is organized as follows. Section 2 introduces preliminary definition and notation, and formally defines the problems of frequency estimation and frequent item detection in data streams. Section 3 recalls related work. ASKETCH is introduced in Section 4. We present ACMSS in Section 5. Experimental results are discussed in Section 6. We draw our conclusions in Section 7.

## 2. Preliminary Definitions

In this Section we briefly recall preliminary definitions and the notation that shall be used throughout the paper. We begin by defining the frequency of weighted items as follows.

**Definition 1.** *Given a stream $\sigma = \{(s_i, w_i)\}_{i=1,2,\dots,n}$ of n pairs (item, weight) with items drawn from the universe $[m] = \{1, 2, \dots, m\}$ and weights which are positive real values, the frequency of an item s is*
$$f_\sigma(s) = \sum_{\substack{i \in [n] \\ s_i = s}} w_i.$$

Next, we define the frequency vector.

**Definition 2.** *A stream $\sigma = \{(s_i, w_i)\}_{i=1,2,\ldots,n}$, whose items are drawn from the universe $[m]$ and whose weights are positive real values, implicitly defines a frequency vector, $\mathbf{f} = (f_1, f_2, \ldots, f_m)$, where $f_i = f_\sigma(i)$ is the frequency of item i.*

We can interpret a stream $\sigma$ as a sequence of updates to the frequency vector $\mathbf{f}$: Initially $\mathbf{f}$ is the null vector, then, for each pair $(i, w)$ in the stream, the entry in $\mathbf{f}$ corresponding to the frequency of the item $i$ is incremented by the corresponding weight $w$.

We are now ready to define the $\epsilon$-approximate frequency estimation problem.

**Definition 3.** *Given a stream $\sigma = \{(s_i, w_i)\}_{i=1,2,\ldots,n}$ of n pairs (item, weight) with items drawn from the universe $[m]$ and weights which are positive real values, the frequency vector $\mathbf{f}$ defined by $\sigma$, and a value $0 < \epsilon < 1$, the $\epsilon$-approximate frequency estimation problem consists in computing a vector $\hat{\mathbf{f}} = (\hat{f}_1, \hat{f}_2, \ldots, \hat{f}_m)$, so that $\left| \hat{f}_i - f_i \right| \leq \epsilon \|\mathbf{f}\|_\ell$, for each $i \in [m]$ where $\ell$ can be either 1 or 2.*

Next, we define $\phi$-frequent weighted items.

**Definition 4.** *Given a stream $\sigma = \{(s_i, w_i)\}_{i=1,2,\ldots,n}$ of n pairs (item, weight) with items drawn from the universe $[m]$ and weights which are positive real values, and a real value $0 < \phi < 1$, the $\phi$-frequent items of $\sigma$ are all those items whose frequency is above $\phi W$, where $W = \sum_{i=1}^{n} w_i$, i.e., the elements in the set $F = \{s \in [m] : f_\sigma(s) > \phi W\}$.*

We will often refer to the $\phi$-frequent items of a stream simply as frequent items, leaving as implicit the reference to a $\phi$ value. Frequent items are also commonly referred to as Heavy Hitters. The $\epsilon$-approximate frequent items problem related to determining $\phi$-frequent weighted items is defined as follows.

**Definition 5.** *Given a stream $\sigma = \{(s_i, w_i)\}_{i=1,2,\ldots,n}$ of n pairs (item, weight) with items drawn from the universe $[m]$ and weights which are positive real values, a real value $0 < \phi < 1$ and a value $0 < \epsilon < \phi$, the $\epsilon$-approximate frequent items problem consists in finding the set F, so that:*

1. *F contains all of the items s with frequency $f_\sigma(s) > \phi W$ where $W = \sum_{i=1}^{n} w_i$ ($\phi$-frequent weighted items);*
2. *F does not contain any item s such that $f_\sigma(s) \leq (\phi - \epsilon)W$.*

## 3. Related Work

The first algorithm for mining frequent items dates back to 1982, and is due to Misra and Gries [18]. Many years later, the LOSSY COUNTING and STICKY SAMPLING algorithms by Manku et al. [19], were published in 2002. Interestingly, in 2003, the Misra and Gries algorithm was independently rediscovered and its computational complexity improved by Demaine et al. [7] and Karp et al. [20]. This algorithm is known in the literature as FREQUENT. Metwally et al. presented a few years later the SPACE-SAVING algorithm [21], which significantly improves the accuracy. These algorithms keep track of frequent items through the use of counters, i.e., data structures managing pair (item, estimated frequency).

Another group of algorithms is based on a sketch data structure, usually a bi-dimensional array hosting a counter in each cell. Pairwise independent hash functions are used to map stream's items to corresponding cells in the sketch. Sketch–based algorithms include COUNTSKETCH by Charikar et al. [5], GROUP TEST [22] and COUNT-MIN [23] by Cormode and Muthukrishnan, HCOUNT [24] by Jin et al. and CMSS [11] by Cafaro et al.

Algorithms for Correlated Heavy Hitters (CHHs) have been recently proposed by Lahiri et al. [25] and by Epicoco et al [26] in which a fast and more accurate algorithm for mining CHHs is presented.

All of the previous algorithms give identical importance to each item. However, in many applications is necessary to discount the effect of old data. Indeed, in some situations recent data

is more useful and valuable than older data; such cases may be handled using the *sliding window* model [27,28] or the time–fading model [29]. The key idea in sliding window is the use of a temporal window to capture fresh, recent items. This window periodically slides forward, allowing detection of only those frequent items falling in the window. In the time–fading model recent items are considered more important than older ones by *fading* the frequency count of older items. Among the algorithms for mining time–faded frequent items we recall $\lambda$-HCOUNT [30] by Chen and Mei, FSSQ (Filtered Space-Saving with Quasi–heap) [31] by Wu et al. and the FDCMSS algorithm [32,33] by Cafaro et al.

Regarding parallel algorithms, Cafaro et al. [34–36] provide parallel versions of the FREQUENT and SPACE-SAVING algorithms for message–passing architectures. Shared-memory versions of LOSSY COUNTING and FREQUENT have been designed by Zhang et al. [37,38], and parallel versions of SPACE-SAVING have been proposed by Dat et al. [39], Roy et al. [40], and Cafaro et al. [41]. Accelerator based algorithms include Govindaraju et al. [42], Erra and Frola [43] and Cafaro et al. [41,44]. Pulimeno et al. [45] present a message-passing based version of the CHHs algorithm [26]; a parallel message-passing based version of [32] is presented in [46].

Distributed mining of heavy hitters include algorithms such as [47–51]. In the context of unstructured P2P networks, gossip–based algorithms have been proposed for mining frequent items including [52–55].

## 4. The ASKETCH Algorithm

In most practical cases where an estimate of the frequency is required, the interesting data is represented by the items prevailing in the stream or of greater weight: In any online shopping site there is the section "best-selling items", on YouTube the section "trends", in Netflix there are three: "The most viewed on Netflix", "the headlines of the moment" and "champions of cashiers". Instead there are very few contexts in which the attention is focused on uncommon or lighter objects. It is for this reason that many algorithms have focused mainly on improving the estimation of frequent items: They are in fact a particular and small group of items whose error strongly affects the accuracy of the produced summaries.

Given the good answers obtained by COUNT-MIN with a smaller space than that required by the other algorithms on sketch, and given the guarantee of overestimation, the research started from this technique and studied the weak points: Collisions with frequent items. The problem is represented both by collisions between two frequent items and those between frequent and infrequent items; if in the first case it is the estimates of both items that are affected, in the second case the difficulty is mainly related to the non frequent item, and the consequent classification error that would make it a false positive. Such errors may seem insignificant if the objective of the analysis is simply to increase sales by highlighting what the public appreciates, but if the estimation of frequency is only an intermediate step, upon which procedures that control thresholds or identify particular statistics must be built, then such errors become crucial.

From COUNT-MIN study it became clear that removing the frequent items from the sketch would lead to a marked improvement in the results, because it would prevent them from colliding with the remaining items, thus eliminating some of the noise. Several approaches have been developed in an attempt to develop optimal techniques to manage and especially to recognize frequent items; among these approaches there is Augmented Sketch, known simply as ASKETCH [10].

ASKETCH is an algorithm that dynamically identifies frequent items and moves them from the main sketch into a structure, called a filter, where no collisions occur. The space is then divided into filter and sketch; the former is made up of a set of $k$ counters shaped as (*item*[$i$], *new_count*[$i$], *old_count*[$i$]), where $i = 1, 2, \ldots, k$ and *item*[$i$] is the item monitored by the $i$-th counter; the sketch can take different forms depending on the frequency estimation algorithm on which ASKETCH rests, in the following we will assume that this is COUNT-MIN. The algorithm is presented as able to solve the

frequency estimation problem but, because of the duplicity of the structure used, it is suitable also for heavy hitter detection.

Let $d'$ and $w'$ be respectively the number of rows and columns of the sketch and let $k$ be the number of filter counters, the *new_count* and *old_count* components are set to zero and the sketch is initialized, as shown in Algorithm 1.

---

**Algorithm 1:** Initialize

---

**Data:** $k$, filter size; $d'$, sketch rows; $w'$, sketch columns
**Result:** filter and sketch initialized
filter is empty;
**for** $i = 1$ *to* $k$ **do**
    *new_count*$[i] = 0$;
    *old_count*$[i] = 0$;
**end**
initialize the sketch;

---

As shown in Algorithm 2, in order to process the pair $(s, v)$ of the stream $\sigma$, ASKETCH checks if $s$ is monitored in one of the filter counters, if so it increases by $v$ the value of the corresponding *new_count* variable, leaving *old_count* unchanged. Otherwise, if there is a free counter in the filter, the item $s$ is stored in that counter by setting *new_count* equal to $v$ and *old_count* equal to zero.

---

**Algorithm 2:** Update

---

**Data:** $(s, v)$, stream pair to be processed
**Result:** filter or sketch updated with new pair $(s, v)$
lookup $s$ in filter;
**if** *item found* **then**
    let $i$ be the index of the counter monitoring $s$;
    *new_count*$[i] \leftarrow$ *new_count*$[i] + v$;
**else if** *filter not full* **then**
    let $i$ be the index of a free counter in the filter;
    *item*$[i] \leftarrow s$;
    *new_count*$[i] \leftarrow v$;
    *old_count*$[i] \leftarrow 0$;
**else**
    update sketch with $(s, v)$;
    $\hat{f}(s) \leftarrow$ CMQUERY$(s)$;
    let $i_m$ be the index of the item with minimum estimated frequency in filter;
    **if** $\hat{f}(s) >$ *new_count*$[i_m]$ **then**
        **if** *new_count*$[i_m] -$ *old_count*$[i_m] > 0$ **then**
            update sketch with $(item[i_m],$ *new_count*$[i_m] -$ *old_count*$[i_m])$;
        **end**
        *item*$[i_m] \leftarrow s$;
        *new_count*$[i_m] \leftarrow \hat{f}(s)$;
        *old_count*$[i_m] \leftarrow \hat{f}(s)$;
    **end**
**end**

---

In case the above conditions are not fulfilled, ASKETCH updates the sketch with $(s, v)$ and estimates the frequency of $s$ using the appropriate Count-Min procedure, denoted by CMQUERY.

Let $\hat{f}(s)$ be the value returned by CMQUERY(s) and $i_m$ be the index of the item in the filter with minimum *new_count* value; if $\hat{f}(s)$ is less than or equal to *new_count*$[i_m]$ no other operation is performed and the next pair is processed. If $\hat{f}(s)$ is strictly greater than *new_count*$[i_m]$, then the item *item*$[i_m]$ is removed from the filter and updated in the sketch with weight given by the difference between *new_count*$[i_m]$ and *old_count*$[i_m]$, if that difference is greater than zero and finally $s$ is put into *item*$[i_m]$ and the related *new_count*$[i_m]$ and *old_count*$[i_m]$ variables are set to the value $\hat{f}(s)$.

The frequency of an item $s$ is given by the *new_count* counter if it is monitored in the filter, otherwise it is computed using the sketch by the Count-Min procedure, as in Algorithm 3.

---

**Algorithm 3:** PointEstimate

---

**Data:** $s$, an item
**Result:** estimation of item $s$ frequency
lookup $s$ in filter;
**if** *item found* **then**
    let $i$ be the index of the counter monitoring $s$;
    **return** *new_count*$[i]$;
**else**
    **return** CMQUERY(s);
**end**

---

The value stored in the *new_count* counter is the real frequency of the item only if it has never been inserted in the sketch, i.e., if *old_count* is equal to zero. In fact, the difference between *new_count* and *old_count* is the exact weight of the item relative to the period of time it remained in the filter. Therefore, as the length of time an heavy hitter item stays in this structure increases, the accuracy of its estimate increases and the possibility of collision with infrequent items decreases. At the same time, if the frequent items are correctly inserted into the filter, the time needed to process the stream is also significantly reduced, since there is no need to determine the frequent items images using the hash functions of the sketch.

ASKETCH can solve, through the Query procedure, the problem of detecting heavy hitters, using $k$ counters in the filter. Obviously the algorithm can generate false negatives if the filter is not sized appropriately; in fact, it can happen that a frequent item $s$ is kept in the sketch because the estimated frequency, although higher than the threshold, does not exceed the value of the minimum *new_count* counter.

It is worth noting here that ASKETCH can also return false positives. In case the input is a static dataset, the problem can be easily solved with a second scan of the dataset, in which the occurrences of all the items reported in the filter at the end of the first analysis are counted.

After swapping items between the two data structures and removing the item *item*$[i_m]$ from the filter, if the difference between *new_count*$[i_m]$ and *old_count*$[i_m]$ is positive, ASKETCH needs to update the counters of the related $d'$ buckets. However, since there is a collision problem in the sketch, once the update is done, it could happen that the value returned by CMQUERY(*item*$[i_m]$) is greater than the new minimum frequency in the filter; then the problem of deciding whether it makes sense to iterate the exchange procedure arises. The authors of ASKETCH have chosen not to make multiple exchanges, but at most one exchange for each processed item, after observing that these can only negatively affect the estimate.

The authors of ASKETCH proved that if multiple exchanges are not allowed, then the maximum number of exchanges between the sketch and the filter is equal to the length of the stream.

## 5. The ACMSS Algorithm

Augmented CMSS, or briefly ACMSS, is our randomized algorithm designed to solve the frequency estimation and the heavy hitter detection problems. Like ASKETCH, its operations are based

on the use of two data structures: A filter and a sketch. The filter is able to correctly monitor the frequency of the elements it manages, for this reason the frequent items are inserted in this structure after being identified as such in the sketch; the two structures therefore communicate dynamically and exchange the management of the items.

A bucket of the sketch keeps track of three variables: Two frequencies and the identity of the candidate as majority item in the bucket. Let $S$ be the sum of the frequencies of all of the items mapped to the bucket by the corresponding hash function; then, the majority item, if it exists, is the item whose frequency exceeds $\lfloor S/2 \rfloor + 1$.

The ACMSS filter consists of $(item[i], count[i])$-shaped counters, with $i = 1, 2, \ldots, k$; $item[i]$ denotes the item monitored by the $i$-th counter and $count[i]$ is its estimated frequency. The sketch, on the other hand, substantially differs from the ASKETCH data structure: Each bucket consists of a tuple of values $(item[i, j], count[i, j], residue[i, j])$ for $i = 1, 2, \ldots, d$ and $j = 1, 2, \ldots, w$, where $d$ and $w$ are respectively the number of rows and columns of the sketch. Through $item[i][j]$ the bucket at row $i$ and column $j$ keeps track of the majority item of its stream, i.e., all of the items falling in that bucket; $count[i][j]$ is an estimation of the frequency of $item[i][j]$, whilst $residue[i][j]$ represents an estimation of the frequency of the other items falling in the bucket. Therefore, by construction, $residue[i][j]$ handles a frequency that is less than or equal to that in $count[i][j]$.

Let $k$ be the number of filter counters, and $\phi$ a fixed support threshold. The size of the sketch is computed, using the approximation parameters $\epsilon$ and $\delta$ and the Formula (2).

Each line of the sketch is associated with a hash function $h_i : [m] \rightarrow [w]$, for $i = 1, 2, \ldots, d$, randomly extracted from a pairwise independent family, where $[m]$ denotes the universe set of the stream. The initialization procedure, shown as Algorithm 4, sets to zero all the counters of both the filter and the sketch. The $W$ value, initialized at zero, is used to keep track of the total weight of the stream that has already been processed.

---

**Algorithm 4:** Initialize

**Data:** $k$, filter size; $d$, sketch rows; $w$, sketch columns; $\phi$, support threshold
**Result:** filter and sketch initialized
filter is empty;
**for** $i = 1$ *to* $k$ **do**
    $count[i] = 0$;
**end**
**for** $i = 1$ *to* $d$ **do**
    **for** $j = 1$ *to* $w$ **do**
        $item[i][j] = 0$;
        $count[i][j] = 0$;
        $residue[i][j] = 0$;
    **end**
**end**
choose $d$ random hash functions $h_1, h_2, \ldots, h_d : [m] \rightarrow [w]$;
set support threshold to $\phi$;
$W \leftarrow 0$.

---

As shown in Algorithm 5, in order to process the $(s, v)$ stream pair, where $s$ is an incoming item and $v$ is its weight, the first data structure to be checked is the filter; if $s$ is already monitored, its estimated frequency is increased by the weight $v$; otherwise, if there is a free counter, the item $s$ is inserted in this structure and the value of the associated frequency counter is equal to $v$. If none of the above conditions occur, the pair is processed by the sketch using the UPDATESKETCH procedure, shown as Algorithm 6.

---

**Algorithm 5:** Update

---

**Data:** $(s, v)$, stream pair to be processed
**Result:** filter or sketch updated with new pair $(s, v)$
$W \leftarrow W + v$;
lookup $s$ in filter;
**if** *item found* **then**
　　let $i$ be the index of the counter monitoring $s$;
　　$count[i] \leftarrow count[i] + v$;
**else if** *filter not full* **then**
　　let $i$ be the index of a free counter in the filter;
　　$item[i] \leftarrow s$;
　　$count[i] \leftarrow v$;
**else**
　　$\hat{f}(s) \leftarrow$ SketchPointEstimate$(s)$;
　　$r \leftarrow$ UpdateSketch$((s, v), \hat{f}(s))$;
　　**if** $r = 1$ **then**
　　　　$\hat{f}(s) \leftarrow \hat{f}(s) + v$;
　　　　let $i_m$ be the index of the item with minimum estimated frequency in filter;
　　　　**if** $\hat{f}(s) > count[i_m]$ **then**
　　　　　　$\hat{f}(item[i_m]) \leftarrow$ SketchPointEstimate$(item[i_m])$;
　　　　　　**if** $count[i_m] - \hat{f}(item[i_m]) > 0$ **then**
　　　　　　　　UpdateSketch$((item[i_m], count[i_m] - \hat{f}(item[i_m])), \hat{f}(item[i_m]))$;
　　　　　　**end**
　　　　　　$item[i_m] \leftarrow s$;
　　　　　　$count[i_m] \leftarrow \hat{f}(s)$;
　　　　**end**
　　**end**
**end**

---

---

**Algorithm 6:** UpdateSketch

---

**Data:** $(s, v)$, stream pair to be processed; $\hat{f}(s)$, estimated frequency of $s$
**Result:** sketch updated with new pair $(s, v)$; $r$, boolean variable worth 1 if $s$ is monitored in at
　　　　least one bucket
$r \leftarrow 0$;
**for** $i = 1$ *to* $d$ **do**
　　$j \leftarrow h_i(s)$;
　　**if** $item[i][j] = s$ **then**
　　　　$count[i][j] \leftarrow$ MAX$(count[i][j], \hat{f}(s) + v)$;
　　　　$r \leftarrow 1$;
　　**else**
　　　　**if** $\hat{f}(s) + v > residue[i][j]$ **then**
　　　　　　**if** $\hat{f}(s) + v > count[i][j]$ **then**
　　　　　　　　$item[i][j] \leftarrow s$;
　　　　　　　　$residue[i][j] \leftarrow count[i][j]$;
　　　　　　　　$count[i][j] \leftarrow \hat{f}(s) + v$;
　　　　　　　　$r \leftarrow 1$;
　　　　　　**else**
　　　　　　　　$residue[i][j] \leftarrow \hat{f}(s) + v$;
　　　　　　**end**
　　　　**end**
　　**end**
**end**
**return** $r$.

---

The update in question is conservative [12,13] since before executing UPDATESKETCH, the frequency of the item $x$ must be estimated using SKETCHPOINTESTIMATE (shown as Algorithm 7).

---

**Algorithm 7:** SketchPointEstimate

**Data:** $s$, an item
**Result:** estimated frequency of item $s$ from sketch
$answer \leftarrow \infty$;
**for** $i = 1$ *to* $d$ **do**
    $j \leftarrow h_i(s)$;
    **if** $item[i][j] = s$ **then**
        $answer \leftarrow$ MIN$(answer, count[i][j])$;
    **else**
        $answer \leftarrow$ MIN$(answer, residue[i][j])$;
    **end**
**end**
**return** *answer*.

---

As in ASKETCH, we evaluate if it is appropriate to pass the management of $s$ to the filter, but, for this to happen, two conditions must be fulfilled: (i) $s$ must be a majority item in at least one bucket of the sketch, i.e., at least one value $item[i][h_i(s)]$ for $i = 1, 2, \ldots, d$ must be equal to the item $s$; (ii) the estimated frequency of $s$ in the sketch, indicated with $\hat{f}(s)$, must be strictly greater than the minimum frequency in the filter. Assuming that both conditions occur and the minimum frequency item in the filter is monitored by the counter with index $i_m$, we update the sketch using the item $item[i_m]$, with weight given by the difference between the value $count[i_m]$ and the output of SKETCHPOINTESTIMATE$(item[i_m])$, when this difference is positive. Then, we assign $s$ to $item[i_m]$ and $\hat{f}(s)$ to $count[i_m]$.

The UPDATESKETCH procedure called on the pair $((s, v), \hat{f}(s))$ performs the conservative update of the sketch for the item $s$ and the weight $v$, taking into account that the current estimated frequency for $s$ is $\hat{f}(s)$.

Let $1 \leq i \leq d$ be the row index, and $j$ the column index of the bucket where $s$ is mapped. In case the value in $item[i][j]$ coincides with $s$ we have to update, only if necessary, the $count[i][j]$ value, so the update happens only if the value in $count[i][j]$ is less than the sum of the $v$ weight and the estimated frequency $\hat{f}(s)$. If, on the other hand, the item in $item[i][j]$ is different from $s$, then the $residue[i][j]$ value should be updated to the new frequency $\hat{f}(s) + v$, taking care of replacing the item stored in $item[i][j]$ if necessary.

So, if updating $residue[i][j]$ is necessary and the new frequency $\hat{f}(s) + v$ is strictly greater than the value of $count[i][j]$, the item monitored by $item[i][j]$ is replaced by $s$, $residue[i][j]$ takes the value $count[i][j]$ and $count[i][j]$ is updated with the frequency $\hat{f}(s) + v$. Note that in the hypothesis just made the update of $residue[i][j]$ is necessary, otherwise it is possible to underestimate the frequency of the items mapped in the bucket that are different from $s$.

The output is the updated sketch and the boolean variable $r$, whose value is 1 if the item $s$ is the majority item in at least one bucket of the sketch.

SKETCHPOINTESTIMATE computes the frequency of the item $s$ considering the minimum value of the estimate obtained scanning the $d$ buckets in which the item is mapped to by the hash functions. Fixing $1 \leq i \leq d$, the frequency of $s$ on the $i$-th row is given by $count[i][h_i(x)]$ if $item[i][h_i(x)]$ is equal to $s$, otherwise it is $residue[i][h_i(x)]$.

The POINTESTIMATE procedure, shown as Algorithm 8, returns the estimated frequency of the item $s$ received as input: We first check if $s$ is monitored in the filter and if so we return the value of $count[i]$, where $i$ is the index of the filter counter monitoring $s$, otherwise SKETCHPOINTESTIMATE is used to estimate the frequency of $s$ in the sketch.

---

**Algorithm 8:** PointEstimate
___

   **Data:** $s$, an item

   **Result:** estimation of item $s$ frequency

   lookup $s$ in filter;

   **if** *item found* **then**

      let $i$ be the index of the counter monitoring $s$;

      **return** $count[i]$;

   **else**

      **return** SKETCHPOINTESTIMATE($s$);

   **end**

---

The QUERY procedure, shown as Algorithm 9, returns the $\mathcal{R}$ set of $\phi$-frequent items with their estimated frequency. $\mathcal{R}$ is, initially, the set of items monitored in the filter whose frequency counter is above the $\phi \cdot W$ threshold. If $c$, which is the number of pairs in $\mathcal{R}$, is less than the filter size, the search for $\phi$-frequent items stops. In fact, in that case, at least one counter, the counter with minimum frequency, is below the $\phi \cdot W$ threshold and we can be sure that all of the frequent candidates monitored by the sketch with frequency above the threshold have already been inserted in the filter.

---

**Algorithm 9:** Query
___

   **Data:** $\phi$, threshold parameter for frequent items; $k$, filter size

   **Result:** set $\mathcal{R}$ of frequent items

   $\mathcal{R} \leftarrow \varnothing$;

   $c \leftarrow 0$;

   **forall** $i$ *in filter* **do**

      **if** $count[i] > \phi \cdot W$ **then**

         $\mathcal{R} \leftarrow \mathcal{R} \cup \{(item[i], count[i])\}$;

         $c \leftarrow c + 1$;

      **end**

   **end**

   **if** $c = k$ **then**

      **for** $i = 1$ *to* $d$ **do**

         **for** $j = 1$ *to* $w$ **do**

            **if** $count[i][j] > \phi \cdot W$ **then**

               $s \leftarrow item[i][j]$;

               $f \leftarrow$ SKETCHPOINTESTIMATE($s$);

               **if** $f > \phi \cdot W$ **then**

                  $\mathcal{R} \leftarrow \mathcal{R} \cup \{(s, f)\}$;

               **end**

            **end**

         **end**

      **end**

   **end**

   **return** $\mathcal{R}$.

---

Instead, if all the monitored items in the filter are in $\mathcal{R}$, then also the minimum counter in the filter has a value above the threshold and there may be some frequent candidates with frequency above the threshold among the items monitored by the sketch. Let $s$ be the element monitored in $item[i][j]$; if the frequency in $count[i][j]$ and the value returned by SKETCHPOINTESTIMATE($s$) are above the $\phi \cdot W$ threshold, then $s$ is inserted into $\mathcal{R}$ with its estimated frequency.

It is worth noting that if the number of counters in the filter is at least $1/\phi$, we never have to search in the sketch. In fact, in that case, the counter with minimum value in the filter is necessarily below the threshold $\phi W$. The reason why is that the sum of all of the counters in the filter, $W_f$, can not be greater than $W$, the total weight of the input stream, for some occurrences are necessarily stored only by the sketch. Then, if the sum $W_f$ is distributed among $k$ counter with $k \geq 1/\phi$ than the counter with minimum value is at most equal to $W_f/k \leq \phi W_f \leq \phi W$.

Now consider a generic item $a$ mapped to the bucket in row $i$ and column $j = h_i(a)$; denoting by $\hat{f}_{i,a}$ the estimate of $a$ obtained in that bucket and with $f(a)$ the real frequency of $a$ it holds that:

$$0 \leq \hat{f}_{i,a} - f(a) \leq residue[i][j]. \tag{1}$$

The first inequality is immediate since ACMSS overestimates the frequency of all items. Moreover, if the item monitored in $item[i][j]$ is different from $a$, then $\hat{f}_{i,a} = residue[i][j]$ and the second inequality holds noting that $f(a) \geq 0$. Otherwise, if the item in $item[i][j]$ coincides with $a$, then $\hat{f}_{i,a} = count[i][j]$, from which the inequality to prove is equivalent to: $count[i][j] - f(a) \leq residue[i][j]$.

Let us first suppose to make a non conservative update, that is to sum the item's weight to all of the buckets in which the item falls: The update is done, in each bucket, exactly as in a summary of the SPACE-SAVING algorithm consisting of two counters, so the inequality is valid [21]. Now consider a conservative update: The error, that is the difference $count[i][j] - f(a)$, decreases further, so the inequality still holds.

**Theorem 1.** *Let $\epsilon$ and $\delta$ denote respectively an error tolerance and a probability of failure; moreover, denote by $\sigma$ a stream of $(item, weight)$ pairs with total weight $W$. Then, the* ACMSS *algorithm making use of a sketch of $d$ rows and $w$ columns, where:*

$$d = \lceil \ln(1/\delta) \rceil, \qquad w = \left\lceil \frac{e}{2\epsilon} \right\rceil, \tag{2}$$

*solves the $(\epsilon, \delta)$-approximate frequency estimation problem for the stream $\sigma$.*

**Proof.** By construction, the error on the frequency estimation of an item can only increase when the item is monitored by the sketch. In fact, during its permanence into the filter, it is tracked exactly. This is the reason why only the size of the sketch affects the bound on the estimation error.

Let $a$ be an item of the universe set $[m]$ managed by the sketch of size $d \times w$; let $i$ be a fixed row index, $j = h_i(a)$ the column index where $a$ is mapped on the $i$-th row by the $i$-th hash function and $X_i = \hat{f}_{i,a} - f(a)$ the variable representing the difference between the estimated frequency on the $i$-th row and the real frequency of $a$. Moreover, let $S_{i,j}$ denote the sum of the frequencies of all the items that fall in the bucket whose row is $i$ and whose column is $j$. Then, using (1), it holds that:

$$0 \leq \hat{f}_{i,a} - f(a) \leq residue[i][j] \leq \frac{S_{i,j}}{2}. \tag{3}$$

Letting $k$ be a generic $[m]$ item, and letting $C_k$ be the indicator random variable associated to the event {the item $k$ is managed by the sketch} and considering the indicator random variable

$$I_{i,j,k} = \begin{cases} 1 & \text{if } h_i(k) = j \\ 0 & \text{otherwise} \end{cases}, \tag{4}$$

it holds that:

$$S_{i,j} = \sum_{k \in [m]} f(k) I_{i,j,k} C_k. \tag{5}$$

Denoting by $W^s$ the total weight managed by the sketch, which is the sum of the weight actually processed by the sketch and the weight processed by the filter but then inserted in the sketch because

of the exchange of items between the two data structures, the expectation of $C_k$ can be approximated with $W^s/W$. The expectation of $S_{i,j}$, considering the uniformity of $h_i$ and the independence of the variables $I_{i,j,k}$ and $C_k$ is:

$$\mathbb{E}[S_{i,j}] = \sum_{k\in[m]} f(k)\mathbb{E}[I_{i,j,k}C_k] = \frac{1}{w}\frac{W^s}{W}\sum_{k\in[m]} f(k) = \frac{1}{w}\frac{W^s}{W}W = \frac{1}{w}W^s. \tag{6}$$

From (3) it follows:

$$\mathbb{E}[X_i] = \mathbb{E}[\hat{f}_{i,a} - f(a)] \leq \frac{1}{2}\mathbb{E}[S_i, j] = \frac{1}{2w}W^s. \tag{7}$$

Applying Markov's inequality to the positive variable $X_i$ we get: $\forall c > 0 \quad \mathbb{P}\left[X_i \geq c\frac{1}{2w}W^s\right] \leq \frac{1}{c}$. Recalling that the algorithm computes the frequency as the minimum estimate obtained on the $d$ rows and exploiting the independence of the variables $X_i$ it holds that:

$$\begin{aligned}
\mathbb{P}\left[\hat{f}(a) - f(a) \geq \frac{c}{2w}W^s\right] &= \mathbb{P}\left[\min_{1\leq i\leq d}\hat{f}_{i,a} - f(a) \geq \frac{c}{2w}W^s\right] \\
&= \mathbb{P}\left[\min_{1\leq i\leq d} X_i \geq \frac{c}{2w}W^s\right] \\
&= \mathbb{P}\left[\bigcap_{i=1}^{d}\left(X_i \geq \frac{c}{2w}W^s\right)\right] \\
&= \prod_{i=1}^{d}\mathbb{P}\left[X_i \geq \frac{c}{2w}W^s\right] \leq \frac{1}{c^d}.
\end{aligned} \tag{8}$$

By choosing $c = e$ and observing that $W^s \ll W$:

$$\mathbb{P}\left[\hat{f}(a) - f(a) < \frac{e}{2w}W\right] \geq \mathbb{P}\left[\hat{f}(a) - f(a) < \frac{e}{2w}W^s\right] \geq 1 - e^{-d}. \tag{9}$$

If $\delta = e^{-d}$ and $\epsilon = e/2w$ we obtain the formula (2) that allow computing the size of the sketch from the required approximation parameters. $\quad\square$

It should be noted that in the previous proof we did not use the conservative update, therefore, taking this into account, the error should be further reduced.

**Theorem 2.** *If l is a ϕ-frequent item and is monitored by the sketch, then l is the majority item in at least one of the d buckets where it is handled with probability greater than or equal to*

$$1 - \left(\frac{1}{2w\phi}\right)^d. \tag{10}$$

**Proof.** Let $l$ be a $\phi$-frequent item mapped in the sketch, $l$ is not recognized as a candidate item to be moved in the filter if, in all $d$ buckets in which $l$ is mapped, the monitored item is different from $l$. The probability of this happening coincides with the probability of the event:

$$\forall 1 \leq i \leq d \quad \hat{f}_{i,l} = residue[i][h_i(l)] \leq \frac{1}{2}S_{i,h_i(l)}, \tag{11}$$

where $\hat{f}_{i,l}$ indicates the estimated frequency of $l$ on the $i$-th row, while $S_{i,h_i(l)}$ represents the sum of the frequencies of all the items falling into the bucket in row $i$ and column $h_i(l)$). However, $l$ is a $\phi$-frequent item and the ACMSS algorithm cannot underestimate the frequency of the items, so the above condition is equivalent to:

$$\forall 1 \leq i \leq d \quad \phi W < f(l) \leq \hat{f}_{i,l} = residue[i][h_i(l)] \leq \frac{1}{2}S_{i,h_i(l)}, \tag{12}$$

where $W$ is the total weight of the stream and $f(l)$ the actual frequency of $l$. Therefore, setting a row index $i$, if $\hat{f}_{i,l} = residue[i][h_i(l)]$ then $\phi W < \frac{1}{2} S_{i,h_i(l)}$ and

$$\mathbb{P}\big[\hat{f}_{i,l} = residue[i][h_i(l)]\big] \leq \mathbb{P}\Big[\phi W < \frac{1}{2} S_{i,h_i(l)}\Big]. \tag{13}$$

Similarly to what we already saw in the previous proof:

$$\mathbb{E}[S_{i,h_i(l)}] = \frac{1}{w} W^s, \tag{14}$$

with $W^s$ the total weight in the sketch. Applying Markov's inequality with $c = 2w\phi$ we get:

$$\begin{aligned}
\mathbb{P}\Big[S_{i,h_i(l)} \geq 2w\phi \cdot \mathbb{E}[S_{i,h_i(l)}]\Big] &= \mathbb{P}\Big[S_{i,h_i(l)} \geq 2w\phi \frac{1}{w} W^s\Big] \\
&= \mathbb{P}\Big[S_{i,h_i(l)} \geq 2\phi W^s\Big] \leq \frac{1}{2w\phi}.
\end{aligned} \tag{15}$$

Moreover, from the inequality $2\phi W > 2\phi W^s$ it follows:

$$\mathbb{P}[2\phi W < S_{i,h_i(l)}] \leq \mathbb{P}[2\phi W^s \leq S_{i,h_i(l)}] \leq \frac{1}{2w\phi}, \tag{16}$$

hence

$$\mathbb{P}\big[\hat{f}_{i,l} = residue[i][h_i(l)]\big] \leq \mathbb{P}\big[2\phi W < S_{i,h_i(l)}\big] \leq \frac{1}{2w\phi}. \tag{17}$$

Therefore, the probability that $l$, a $\phi$-frequent item, is not monitored in the bucket corresponding to row $i$, and column $h_i(l)$ is at most $1/2w\phi$. The item $l$ is not a candidate to be moved in the filter if it is not monitored in any bucket, and the probability of this happening is:

$$\mathbb{P}\Big[\bigcap_{i=1}^{d} (\hat{f}_{i,l} = residue[i][h_i(l)])\Big] \leq \Big(\frac{1}{2w\phi}\Big)^d, \tag{18}$$

being the rows of the sketch independent.    □

As a final remark, the following Lemma holds, the proof of which follows straight from the algorithm's operations.

**Lemma 1.** *Let $l$ be a $\phi$-frequent item; $l$ is not reported by the* QUERY *procedure of* ACMSS *and therefore it is a* false negative *if and only if it is not monitored in the filter and it is not a majority item in any bucket of the sketch.*

For what stated by Theorem 2 and Lemma 1 we can be confident that the probability of a $\phi$-frequent item being a false negative for ACMSS is indeed very low. Furthermore, Theorem 1 provides assurances on the error in frequency estimation committed by the algorithm. Therefore, we can state that, with a high probability which depends on the size of the data structures used, ACMSS also solves the $\epsilon$-approximate frequent items problem (Definition 5).

## 6. Experimental Results

We shall compare the experimental results, obtained on synthetic datasets, of the ACMSS and ASKETCH algorithms.

The hash functions used to manage the data structures are the same for all the algorithms. In particular, for the rows of the sketch the $xxHash$ function has been chosen; instead, the filter has

been implemented through a SPACE-SAVING Stream-Summary and the *hash*31 function has been used to arrange the items in the hash table.

The algorithms have been implemented in C++; the source code was compiled using the Intel c++ v19.0.4 compiler on linux CentOS 7 with the following flags: -O3 -std=c++14. The tests were performed on a workstation with 64 GB RAM and two 2.0 GHz Intel Xeon exa-core E5-2620 CPUs with 15 MB cache level 3.

The metric used to evaluate the efficiency of the algorithms is stream processing throughput, corresponding to the average number of pairs (items, weight) processed per time unit and measured in items per millisecond.

In order to evaluate the accuracy of the algorithms with regard to the frequency estimation problem, the following metrics have been used:

- average absolute error, given by the sum, obtained on all the items of the whole universe $[m]$, of the difference between estimated frequency and real frequency of the item, divided by the cardinality of the whole $[m]$: $AverageAbsoluteError = \frac{\sum\limits_{i \in [m]} |\hat{f}(i) - f(i)|}{m}$;

- absolute maximum error, i.e., the maximum variation between estimated and real frequency, recorded on all items in the universe as a whole;

- average relative error which is given by the ratio between the sum of relative errors on items with positive frequency and the number of items that satisfy the latter condition. The relative error of an item is the difference between the estimated and the actual frequency, divided by its exact frequency: $AverageRelativeError = \frac{\sum\limits_{i \in [m]:\ f(i) > 0} |\hat{f}(i) - f(i)| / f(i)}{|i \in [m]:\ f(i) > 0|}$;

- maximum relative error, i.e., the maximum relative error obtained on items whose frequency is positive.

The precision and recall metrics instead allow evaluating the accuracy with regard to the problem of determining the heavy hitters.

In order to obtain a fair comparison, exactly the same amount of memory was used for the two algorithms; in particular the number of rows $d$ of all sketches was set to 4, as this value corresponds to a very small $\delta$ probability of failure.

The number of filter counters, denoted by $k$, has also been set to 32 for both algorithms. The authors of ASKETCH have in fact observed that this value represents a threshold: A further increase of the filter size would lead to a reduction of $filter_{selectivity}$, that is the ratio between the weight processed by the sketch and the total weight of the stream, really negligible, while it would continue to increase the error in the sketch due to the collisions.

Considering finally $w$, which is the number of columns used in ASKETCH, the value of the corresponding parameter for ACMSS, denoted by $w'$, has been obtained using the formula $w' = 2w/5 + 2k/5d$.

The values of $w$ and $w'$ used in the tests are listed in Table 1. The ASKETCH QUERY procedure used to detect heavy hitters has been slightly modified with regard to the corresponding pseudo-code reported in [10]. In fact, the function has been implemented in order to return only the items contained in the filter with estimated frequency strictly higher than $\phi W$, where $W$ represents the total weight of the stream. In this way QUERY works similarly to the corresponding function of the other algorithm.

**Table 1.** Number of columns used.

| $w$ (ASKETCH) | 250 | 500 | 750 | 1000 | 1250 |
|---|---|---|---|---|---|
| $w'$ (ACMSS) | 103 | 203 | 303 | 403 | 503 |

The synthetic datasets, on which the tests were conducted, have a zipfian distribution; the items are represented by unsigned int variables requiring 32 bits, while the associated weights have all been fixed, without loss of generality, to 1.

The seed, used for the pseudo-random number generator associated to the distribution, has been varied between 10 possible values, taking care to use in each experiment the same identical value for both the algorithms to test.

The comparison between the algorithms was conducted by varying the number $w$ of columns of ASKETCH and then the amount of memory occupied (budgeted memory), the skew $\rho$ parameter of the distribution and the support threshold $\phi$, as reported in Table 2. So, for each different value of $\rho$ and $\phi$, the tests were performed 10 times as the seed varied and the results were derived from the test average. The length of the streams and the number of possible distinct items were fixed at $10^7$.

**Table 2.** Parameter values.

| Parameters | Values | Default Value |
| --- | --- | --- |
| w | 250, 500, 750, 1000, 1250 | 500 |
| $\rho$ | 1.0, 1.1, 1.3, 1.7, 2.5 | 1.3 |
| $\phi$ | 0.0005, 0.001, 0.002, 0.004, 0.008 | 0.002 |

The total space occupied by the algorithms has been varied between 8640 (in case $w = 250$) and 40,640 bytes (in case $w = 1250$). The default value of $w$ has been set to 500 and consequently the default space is 16,640 bytes.

As shown in Figure 1 the ASKETCH algorithm is more efficient than the ACMSS algorithm. In fact if an item $i$ is processed in the sketch and if, after the update, this item must be evicted from the sketch and moved to the filter, ACMSS performs more work than ASKETCH. If $x$ is the item to be removed from the filter, ACMSS must call SKETCHPOINTESTIMATE($x$) to know how much to increase the buckets counters associated with $x$; on the contrary ASKETCH avoids the call, since this information is contained in *old_count*[$x$]. It should be noted that the gap between ASKETCH and ACMSS processing speed, remains constant as the amount of memory available to the algorithms varies.

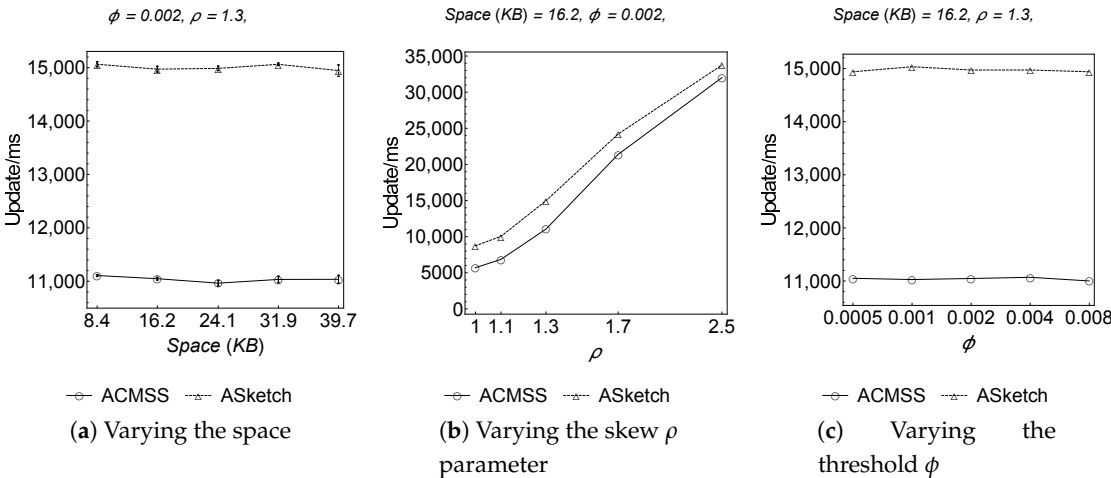

**Figure 1.** Efficiency of the algorithms: Stream processing throughput.

On the other hand, Figure 2 and Table 3 make it clear that the Average Absolute Error committed by ACMSS, especially on a small memory space, is lower than that committed by ASKETCH. Even wider is the gap between the Absolute Maximum Error of these algorithms; therefore, although the columns of the ACMSS sketch are less than half of those of ASKETCH (see Table 1),

the strategy of allocating a Space-Saving mini-summary in each bucket of the sketch produces an increase in the accuracy of the estimate compared to the competitor.

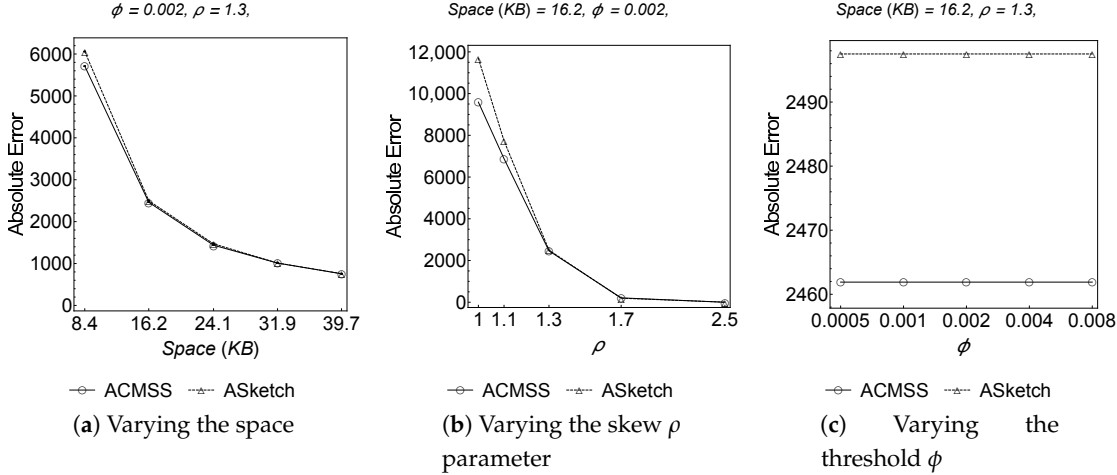

**Figure 2.** Accuracy of the algorithms: Average Absolute Error.

**Table 3.** Absolute Error (average and maximum), zipfian distribution with skew $\rho = 1.3$ and support threshold $\phi = 0.002$.

| Space (KB) | 8.44 | 16.25 | 24.06 | 31.87 | 39.69 |
|---|---|---|---|---|---|
| Absolute Error: Average (ASketch) | 6059.87 | 2497.51 | 1470.74 | 1011.89 | 759.34 |
| Absolute Error: Average (ACMSS) | 5710.35 | 2461.90 | 1437.89 | 1010.71 | 754.91 |
| Absolute Error: Maximum (ASketch) | 36,061 | 30,228 | 21,596 | 15,824 | 17,234 |
| Absolute Error: Maximum (ACMSS) | 17,606 | 11,018 | 6346 | 6031 | 4913 |

The remarks made for the Absolute Error apply similarly to the Relative Error, as shown in Figure 3 and Table 4. As shown in Figure 4 and Table 5, the ACMSS recall is always equal to 100% unless the $\phi$ threshold is very small; however, even with a $\phi$ threshold = 0. 0005 ACMSS obtains a recall of 99.34%, so there is almost no chance that this algorithm produces false negatives. The excellent results of ACMSS stem from the possibility of continuing the search for heavy hitters in the sketch, and not just in the filter.

**Table 4.** Relative Error: Average and maximum, zipfian distribution with skew $\rho = 1.3$ and support threshold $\phi = 0.002$.

| Space (KB) | 8.44 | 16.25 | 24.06 | 31.87 | 39.69 |
|---|---|---|---|---|---|
| Relative Error: Average (ASketch) | 4841.65 | 1993.86 | 1175.07 | 807.61 | 606.38 |
| Relative Error: Average (ACMSS) | 4558.88 | 1964.95 | 1147.28 | 806.20 | 601.94 |
| Relative Error: Maximum (ASketch) | 31,554.00 | 20,760.00 | 16,175.00 | 10,787.00 | 12,337.00 |
| Relative Error: Maximum (ACMSS) | 11,028.00 | 6974.00 | 4864.00 | 3465.00 | 3082.00 |

**Table 5.** Recall, zipfian distribution with skew $\rho = 1.3$ and budgeted memory equal to 16.25 KB.

| $\phi$ | 0.0005 | 0.001 | 0.002 | 0.004 | 0.008 |
|---|---|---|---|---|---|
| Recall ASketch | 26.40 | 45.26 | 77.86 | 100 | 100 |
| Recall ACMSS | 99.34 | 100 | 100 | 100 | 100 |

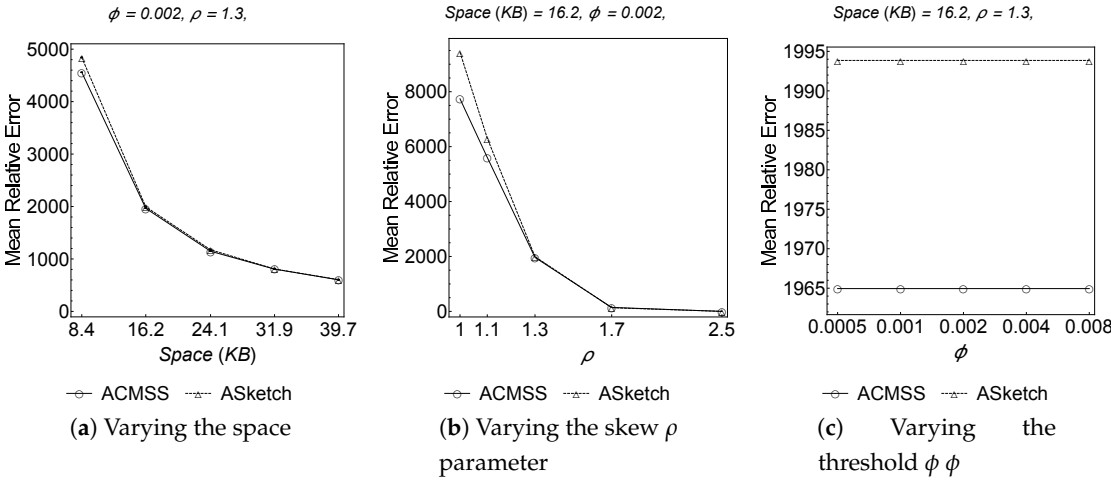

**Figure 3.** Accuracy of the algorithms: Average Relative Error.

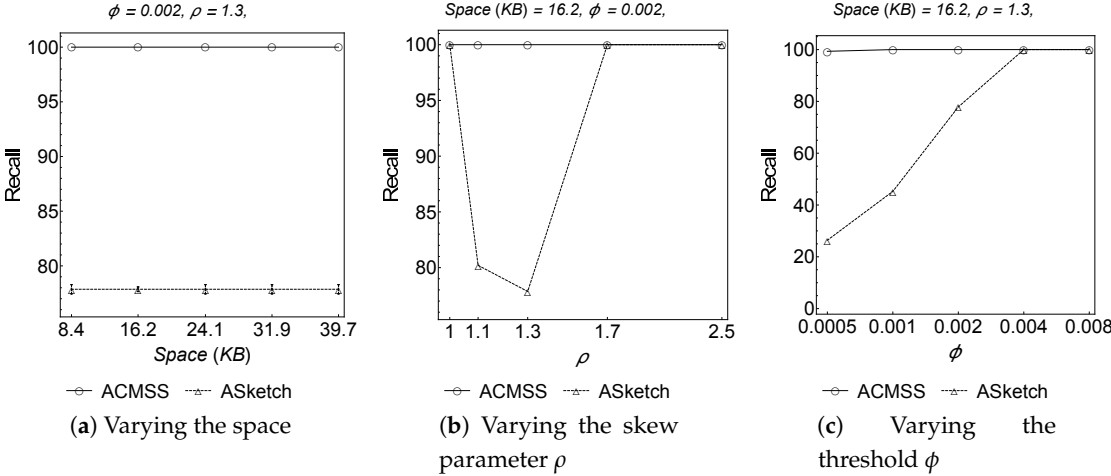

**Figure 4.** Accuracy of the algorithms: Recall.

In Figure 5 it can be finally observed that our algorithm, when all the parameters considered vary, obtains 100% of precision, so each item detected as frequent actually is a frequent item.

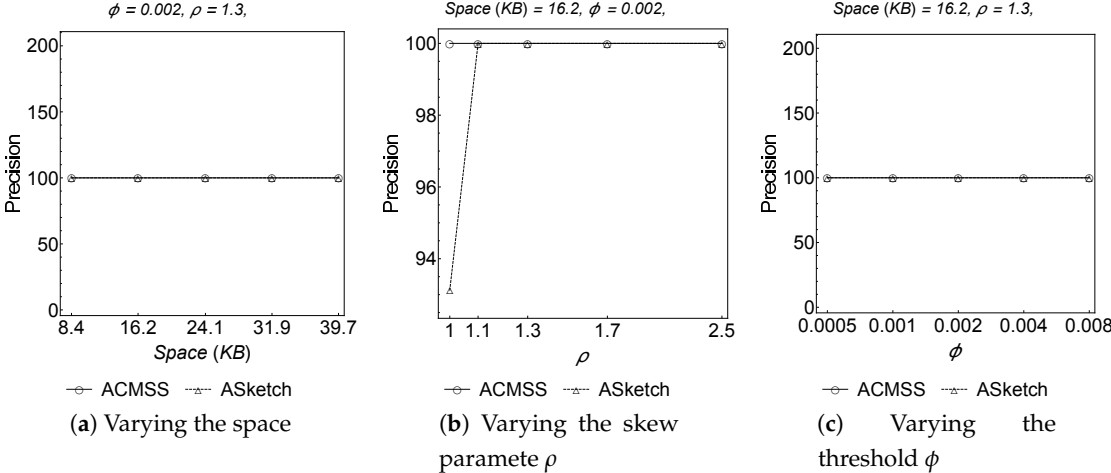

**Figure 5.** Accuracy of the algorithms: Precision.

## 7. Conclusions

We designed and analyzed ACMSS, an algorithm for frequency estimation and heavy hitter detection, and compared it against the state of the art ASKETCH algorithm. Through extensive experimental results we have shown that, given the same budgeted amount of memory, for the task of frequency estimation our algorithm outperforms ASKETCH with regard to accuracy. Moreover, regarding the task of heavy hitter detection, we have also shown that, under the assumptions stated by its authors, ASKETCH may not be able to report all of the heavy hitters, whilst ACMSS will always provide with high probability the full list of heavy hitters.

**Author Contributions:** Conceptualization, M.P., M.C. and I.E.; methodology, F.V., M.P., M.C. and I.E.; software, F.V., M.P., M.C. and I.E.; validation, M.P., M.C. and I.E.; formal analysis, F.V., M.P., M.C. and I.E.; investigation, M.P., M.C. and I.E.; resources, M.C. and I.E.; data curation, M.P., M.C. and I.E.; writing—original draft preparation, M.C.; writing—review and editing, F.V., M.P., M.C. and I.E.; visualization, I.E.; supervision, M.C. and I.E.; project administration, M.C. and I.E. All authors have read and agreed to the published version of the manuscript.

**Funding:** This research received no external funding.

**Conflicts of Interest:** The authors declare no conflict of interest.

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
