# Peer review of "On Frequency Estimation and Detection of Heavy Hitters in Data Streams"

_futureinternet, doi:10.3390/fi12090158_

Round 1

Reviewer 1 Report

In this paper the authors design and analyze ACMSS, an algorithm for frequency estimation and heavy hitter detection, and compare it against the state of the art ASKETCH algorithm. Experimental study is conducted to show that their algorithm outperforms ASKETCH with regard to accuracy under certain conditions.

The paper is interesting. The main concern is the contribution of the paper. Although the ACMSS algorithm is more accurate than the ASKETCH algorithm, the ASKETCH algorithm is more efficient than the ACMSS algorithm. The authors should find and emphasize the applications where the accuracy is more important while running time is tolerable.

It is not clear about the innovation design in ACMSS that makes it outperform ASKETCH with regard to accuracy. The authors may highlight it in the introduction.

Some minor error/confusing: line 155 says “where the frequency is monitored correctly”; line 179 says “may not be the real frequency of the item”.

Reviewer 2 Report

In this paper, Authors designed and analyzed ACMSS for frequency estimation and heavy hitter detection. In addition, it is also compared with the state of the art ASKETCH algorithm. I appreciate authors for producing such a valuable research work. The paper is presented well, and everything can be easily readable and understandable. I have just only one suggestion, An advanced methods have been proposed for data streams handling in different domains. Therefore, authors should also mention those methods, which may help readers to know how to handle data streams in different domain (“An intelligent healthcare monitoring framework using wearable sensors and social networking data” and “Strategies for data stream mining method applied in anomaly detection” )

Round 2

Reviewer 1 Report

I'm very glad that the authors revise the paper as suggested.